# OpenReview forum: "Bridging The Gap between Low-rank and Orthogonal Adaptation via Householder Reflection Adaptation"
_NeurIPS.cc/2024/Conference — NeurIPS 2024 spotlight_

### Official Review · Reviewer_mtnZ · 2024-06-30

**Soundness:** 3
**Presentation:** 3
**Contribution:** 3
**Rating:** 5
**Confidence:** 4

**Summary:**

Both low-rank and orthogonal adaptation techniques can effectively adapt large-scale pre-training models to downstream tasks. This work proposes an new adaptation methods based on Householder reflections(HR). It discloses the connection between low-rank and orthogonal adaptation and builds a unified adapter-based fine-tuning framework. the proposed method can further save the amount of learnable parameters and achieve superior performance when compared to existing methods.

**Strengths:**

The disclosed relationship between low-rank and orthogonal adaptation is meaningful in building a unified adapter-based fine-tuning framework. The HRA technique saves a lot of learnable parameter，the experimental results are strong.

**Weaknesses:**

The structure of the paper is not well developed. The abstract and the introduction need some revision, the motivation for building a unified framework is not clear, what’s the pros and corns of the low-rank and orthogonal adaptation techniques? What kind of advantages are they trying to combine from both techniques? The authors mentioned the gap between the two techniques, does this gap have a negative effect on the adaptation?

**Questions:**

1. Section 3，an overview about the motivation, novelty and organization is needed at the beginning of section 3. It is rambling to start with the details of the proposed method.

2. In the experiments, the parameter r is empirically set to some fixed value, please provide some advices on setting r for real application.

3. What about the training cost and inference time of the proposed method when compared to the others?

4. In section 4.3, which part of the model is applied with the proposed HRA? Does each layer has a similar r?

5. citation 33 is published in 2023.

**Limitations:**

no. please talk about the risk of using such models in real applications, also the potential negative societal impact.

---

> ### Author Rebuttal · Authors · 2024-08-06
>
> Thanks for your appreciation of our work and constructive comments. Below, we try to resolve your concerns one by one.
>
> **Q1: Improve the structure of the paper and highlight the motivation for building a unified adaptation framework.**
>
> **A1:** We have detailed the pros and cons of LoRA and OFT in the Related Work section. Regarding their advantages, LoRA assumes that weight changes during model adaptation have a low "intrinsic rank," while OFT preserves pre-trained knowledge by maintaining the pairwise angles between neuron vectors. As for their drawbacks, LoRA cannot ensure the preservation of angles between neuron vectors, and OFT can only achieve low-rank weight updates when $(R−I)$ is low-rank. HRA combines these two strategies, leveraging their advantages jointly while suppressing their drawbacks simultaneously.
>
> Moreover, building a unified adaptation framework is insightful for revisiting the recent rapid development of various adaptation methods, which may help inspire new technical routes. We have placed an overview of the motivation, novelty, and organization at the end of the Introduction. We plan to polish our paper in the final version further. Thanks for your suggestion.
>
> **Q2: How to set r for real applications?**
>
> **A2:** **We have discussed the setting of $r$ in the above general response, which may have resolved your concern to some extent.** In this study, we set $r\leq8$ for most datasets and tasks because this setting is sufficient to achieve superior adaptation results with fewer trainable parameters. Therefore, we empirically recommend starting with $r=8$. If the model performance is not satisfactory, increasing $r$ will usually work better. As we mentioned in the above general response, like AdaLoRA, we will consider adjusting the $r$ of HRA adaptively as our future work, but it is not the main contribution of this paper.
>
> **Q3: The training and inference time of HRA.**
>
> **A3:** **We have shown the training time and computational efficiency of HRA in the general response.** After training, we multiply the learned orthogonal matrices with the weight matrices, leading to a new model without increasing parameters. Therefore, like LoRA and OFT, HRA does not change the model's inference time.
>
> **Q4: In section 4.3, which part of the model is applied with the proposed HRA? Does each layer have a similar r?**
>
> **A4:** We follow the experimental settings of LoRA and OFT for a fair comparison. For the stable diffusion model, we apply HRA to its attention modules. In this study, we apply the same $r$ to the weight matrices of each attention module. Note that such a simple setting has resulted in superior performance. In the future, we can leverage the same idea of AdaLoRA, adjusting $r$ for different layers.
>
> **Q5: Reference about OFT. Citation 33 was published in 2023.**
>
> **A5:** Thank you for pointing out this mistake. We will correct it in the revised paper.

---

> > ### Comment · Reviewer_mtnZ · 2024-08-13
> >
> > Thank you for the rebuttal. It addresses all of my concerns from the initial review.

---

> > > ### Author Response · Authors · 2024-08-13
> > >
> > > Thanks for your response. It would be nice if our responses helped evaluate our work further and lead to a higher final score.

---

### Official Review · Reviewer_1vEG · 2024-07-09

**Soundness:** 3
**Presentation:** 4
**Contribution:** 3
**Rating:** 7
**Confidence:** 4

**Summary:**

his paper proposes a simple yet efficient adaption method, namely HRA, which finetunes a pretrained model by multiplying each frozen weight matrix with an orthogonal matrix constructed by a chain of learnable Householder reflections. The authors interpret HRA as an adaptive LoRA that retains theoretical guarantees of preserving pre-training knowledge of OFT, somewhat bridging the gap between LoRA and OFT. Number of trained parameters and computational complexity are analyzed. Experiments on several pretrained models (DeBERTa, LLaMA2, Stable Diffusion) show that HRA with fewer learnable parameters and suitable orthogonality regularizing strength achieves superior performance performances than existing methods, demonstrating the effectiveness of HRA for different downstream tasks.

**Strengths:**

This paper is well motivated and well written.

The authors theoretically show that HRA can be formulated as an adaptive LoRA, providing a new perspective that bridges OFT to LoRA, which is insightful.

To show the effectiveness and wide applicability of HRA, this paper has fairly conducted various types of experiments on different tasks including traditional NLP tasks GLUE, LLM tasks GSM8K/MATH, and multimodal tasks text2image generation.

**Weaknesses:**

This paper claims that HRA inherits the theoretical guarantee of OFT on the retention of pretraining knowledge. However, this paper seems a bit overclaim about this without experiments back it up.

**Questions:**

What are the limitations of HRA?
How is the wall clock cost, comparing HRA to LoRA during training? Could this paper show the time cost comparing HRA to LoRA?

**Limitations:**

I would like to see the authors to discuss on the limitations of the proposed method.

---

> ### Author Rebuttal · Authors · 2024-08-06
>
> Thanks for your appreciation of our work. **We have resolved your concerns about the limitations and the computational efficiency of HRA in the above general response.** For your remaining concerns, we provide our answer below.
>
> **Q1: The evidence on the retention of prior knowledge.**
>
> **A1:** Thanks for your constructive suggestion. To verify our claim, we fine-tune LLaMA-2 7B on the MATHQA dataset by LoRA and HRA, respectively, and check the degradation of model performance on classic NLP tasks, including typical language tasks in ARC, HellaSwag, MMLU, and Winogrande, and a coding task in HumanEval. For a fair comparison, we apply the same number of trainable parameters and the same batch size for LoRA and HRA. Ideally, after adaptation, we hope that the model can still maintain its high performance in the NLP tasks.
>
> | Model | ARC | HellaSwag | MMLU | Winogrande | HumanEval |
> | :-----| :----: | :----: | :----: | :----: | :----: |
> | LLaMA2 7B | 49.74 | 58.90 | 45.92 | 74.11 | 12.80 |
> | LLaMA2 7B fine-tuned by LoRA | 48.81 | 56.89 | 40.60 | 71.27 | 11.59 |
> | LLaMA2 7B fine-tuned by HRA | 49.57 | 57.72 | 41.20 | 73.32 | 13.41 |
>
> According to the above results, we find that compared to LoRA, HRA retains more of the original model's knowledge, whose performance degradation is less severe than LoRA's. In the HumanEval task, its performance is even better than that of the original model (which we think is because the MATHQA dataset contains many samples relevant to logic and reasoning tasks and thus is useful in the HumanEval task).
>
> We will add this experimental result in the final paper. Thanks again for your suggestion.

---

> > ### Comment · Reviewer_1vEG · 2024-08-13
> > **Response to Rebuttal**
> >
> > Thanks to the author's for their responses. I am glad to see the additional results and more confident to vote acceptance for this paper.

---

> > > ### Author Response · Authors · 2024-08-13
> > >
> > > Thanks for your appreciation of our work. We will include additional experimental results and corresponding analyses in the final version of the paper.

---

### Official Review · Reviewer_7LSp · 2024-07-12

**Soundness:** 3
**Presentation:** 3
**Contribution:** 3
**Rating:** 5
**Confidence:** 4

**Summary:**

The paper proposes a simple but effective adaptation method based on Householder reflection matrix. The authors show that this method is closely related to low-rank adaption. Diverse experiments demonstrated the effectiveness of the proposed method in comparison to a few baselines.

**Strengths:**

1. The idea of using Householder reflection matrix to do deep learning mode adaption is novel.
2. The proposed method is simple and effective.
3. The experiments are extensive.

**Weaknesses:**

1. The proposed method is a special case of Orthogonal Fine-Tuning. The major difference is the use of the Householder reflection matrix. This means the novelty is relatively limited.
2. The paper does not explain why the proposed method outperforms baselines such as LoRA and OFT.
3. In Figure 1(c), the table, $\lambda=1e-4$ is better than $\lambda=\infty$ and $0$. The authors should report the performance of other values of $\lambda$ such as $1e-1$ to $1e-8$ to show the impact of $\lambda$.
4. According to Figure 1(c), it seems that the gain of the proposed method is many from the regularization. This raises a question---will orthogonal regularizations also improve LoRA and OFT? The authors should make a fair comparison and show the source of the improvement of the proposed method clearly.

**Questions:**

Please refer to the weaknesses.

**Limitations:**

The reviewer hasn't found the discussion of limitations.

---

> ### Author Rebuttal · Authors · 2024-08-06
>
> Thanks for your comments. Below we try to resolve your concerns one by one.
>
> **Q1: The novelty of the proposed method.**
>
> **A1:** We respectfully disagree with the comment that our work's novelty is relatively limited for the following three reasons. Firstly, to our knowledge, our work makes the first attempt to bridge the gap between low-rank and orthogonal adaptation techniques. Our HRA is a new implementation of OFT. At the same time, it is also an adaptive LoRA, as we mentioned in Section 3.3. Secondly, existing OFT restricts the interactions between different dimensions of the weight matrix due to the block diagonal structure of its orthogonal matrix, while BOFT overcomes this issue by time-consuming butterfly matrix multiplication. Unlike these two methods, our HRA implements a chain of Householder reflections to implement orthogonal adaptation, as discussed in Section 3.2. This implementation leads to better efficiency and performance. Thirdly, focusing on HRA, we analyze the impact of the orthogonality of reflection planes on the adaptation performance (Section 3.4), proposing an orthogonal regularizer in the training phase, which can further boost the adaptation performance.
>
> We believe all three of the above contributions are new to the research community, and thus, the novelty of our work is sufficient. Actually, **in your first comment in the Strengths session, you admitted the novelty of our work.** Therefore, we hope that the above response helps you reconsider our work.
>
> **Q2: The reasons for the superiority of HRA compared with LoRA and OFT are not explained.**
>
> **A2:** In fact, we have explained the reasons for the superiority of HRA. In particular, in Section 3.2, we compared HRA with OFT and BOFT on their implementations and computational complexity. **As shown in Lines 147-148 and 157-158, we have shown that HRA can be more efficient (i.e., using fewer trainable parameters) in a mild condition, which is easy to meet in our experiments.**
>
> Regarding the retention of pre-training knowledge, **we have shown in Lines 168-172 that HRA can preserve the angular information of weight matrices like OFT and BOFT do,** which is better than LoRA.
>
> In addition, by introducing an orthogonal regularizer during adaptation, we control the orthogonality of reflection planes and thus help achieve a trade-off between the model capacity and regularity. Compared to LoRA and OFT, this mechanism can further reduce the risk of over-fitting, which helps further boost our performance. We plan to add this point to the end of section 3.4.
>
> **Q3: More experimental results with other values of $\lambda$.**
>
> **A3:** We conduct the experiments with other values of $\lambda$, and the results are as follows.
>
> | Method | #Param | GSM8K | MATH |
> | :-----| :----: | :----: | :----: |
> | $\text{LoRA}_{r=32}$ | 0.25% | 50.2 | 7.8 |
> | $\text{OFT}_{b=16}$ | 0.13% | 50.1 | 8.4 |
> | $\text{HRA}_{r=32,\lambda=\infty}$ | 0.12% | 52.8 | 9.2 |
> | $\text{HRA}_{r=32,\lambda=1e-1}$ | 0.12% | 53.6 | 8.3 |
> | $\text{HRA}_{r=32,\lambda=1e-4}$ | 0.12% | 56.3 | 9.3 |
> | $\text{HRA}_{r=32,\lambda=1e-8}$ | 0.12% | 53.6 | 8.6 |
> | $\text{HRA}_{r=32,\lambda=0}$ | 0.12% | 55.8 | 9.0 |
>
> We can find that a) the performance of HRA is relatively stable concerning the change of $\lambda$, b) in the wide range of $\lambda$, HRA is superior to the baselines, c) even if ignoring the regularizer $(\lambda=0$), our method still outperforms the baselines, which demonstrates the effectiveness of implementing orthogonal adaptation based on Householder reflections.
>
> **Q4: Will the orthogonal regularizer also improve LoRA and OFT?**
>
> **A4:** This is an interesting question. Firstly, it should be emphasized that **our comparison experiments are fair because, in all Tables and Figures, we have considered the HRA with $\lambda=0$ (i.e., without the orthogonal regularizer) and compare it with the baselines.** The experimental results show that HRA can outperform LoRA, OFT, and other competitors even without the regularizer. **In other words, the superiority of our method is mainly from the proposed Householder reflection chain, and the proposed regularizer can further boost performance.**
>
> Secondly, it is nonsense to apply the orthogonal regularizer to OFT because the block diagonal structure of its orthogonal parameter matrix has ensured that the columns of the matrix are orthogonal to each other. For BOFT, the columns of different butterfly orthogonal parameter matrices are also orthogonal to each other. In other words, **OFT and its variants already have intrinsic and strict orthogonality constraints**.
>
> Finally, for the vanilla LoRA in the formulation $W+AB$, we can impose our orthogonal regularizer on its parameter matrices, which was never considered before. Although our regularizer is motivated by the orthogonality of reflection planes rather than LoRA, to resolve your concern, we imposed our regularizer on the matrix $B$ of LoRA and tested it in the mathematical reasoning task.
>
> | Method | #Param | GSM8K | MATH |
> | :-----| :----: | :----: | :----: |
> | $\text{LoRA}_{r=16}$ | 0.12% | 47.3 | 6.6 |
> |  $\text{LoRA}_{r=16,\lambda=1e-4}$ | 0.12% | 47.7 | 6.8 |
> |  $\text{HRA}_{r=32,\lambda=0}$ | 0.12% | **55.8** | **9.0** |
> |  $\text{HRA}_{r=32,\lambda=1e-4}$ | 0.12% | **56.3** | **9.3** |
>
> The above results verify our claim --- the superiority of HRA is mainly caused by the Householder reflection chain. **Note that, without a strong theoretical motivation like the orthogonality of reflection planes, we have too many options to combine LoRA with our regularizer**, e.g., imposing our regularizer to $A$, $B$, $AB$, and so on. Considering so many variants of LoRA is out of the scope of this work.
>
> We hope the above responses can resolve your concerns. We are willing to discuss with you in the next phase if you have any other questions.

---

> > ### Comment · Reviewer_7LSp · 2024-08-13
> >
> > I appreciate the rebuttal. I have raised the rating to 5.

---

> ### Author Response · Authors · 2024-08-13
>
> We are grateful that our response has positively impacted the rating. If you have any further comments or questions, we would be happy to address them.

---

### Official Review · Reviewer_dePa · 2024-07-12

**Soundness:** 3
**Presentation:** 3
**Contribution:** 3
**Rating:** 7
**Confidence:** 3

**Summary:**

This paper proposed a new model fine-tuning method, called Householder reflection adaptation (HRA). The main idea of HRA is to fine-tune the model with a series of Householder reflections. By virtue of the Householder reflection, the orthogonality of the tuning matrix can be obtained, the number of tuning parameters can be reduced, and the computational cost can also be saved. Besides, with simple math, it can be shown that HRA also share the low-rank property with LoRA. Experiments on several tasks have been conducted, to demonstrate the advantages of the proposed method.

**Strengths:**

The idea is simple, yet very effective. Empirical results show that the proposed HRA can achieve better fine-tuning results with less trainable parameters.

**Weaknesses:**

I do not find major flaws of this work.

**Questions:**

I have no further questions.

**Limitations:**

The authors do not explicitly discuss the limitations and broader societal impacts of this work. Since the proposed method could be applied to LLMs, which have been used by more and more people, the broader societal impacts should be discussed.

---

> ### Author Rebuttal · Authors · 2024-08-06
>
> Thanks for your appreciation of our work. We believe your concerns have been resolved in our general response, and we hope that our response can help increase your confidence score. We are willing to discuss with you in the next discussion phase if you have any other questions.

---

### Author Rebuttal · Authors · 2024-08-06

We thank all the reviewers for their appreciation of our work. Below, we provide a general response to their common concerns and a specific response to each reviewer's remaining questions.

**Q1: Discussions on limitations and societal impacts of this work.**

**A1:** Regarding the limitations of HRA, we believe the main concern is the setting of hyperparameters (i.e., the rank $r$ and the weight of orthogonal regularizer $\lambda$). Similar to LoRA, the rank $r$ of our HRA determines the trade-off between the number of trainable parameters and the training efficiency. In this study, we set $r$ to ensure that the number of our trainable parameters is smaller than those of baselines. Of course, inspired by the recent variants of LoRA, e.g., AdaLoRA, we can adjust the rank $r$ adaptively, which is not the main contribution of this work and thus is left to be our future work.

For the weight $\lambda$, which determines the trade-off between the expressiveness and the regularity of the adapter, we set it in a range to quantitatively analyze the impact of orthogonality. In particular, following the suggestion of Reviewer 7LSp, we conducted mathematical reasoning experiments with different values of $\lambda$, and the results are as follows.

| Method | #Param | GSM8K | MATH |
| :-----| :----: | :----: | :----: |
| $\text{LoRA}_{r=32}$ | 0.25% | 50.2 | 7.8 |
| $\text{OFT}_{b=16}$ | 0.13% | 50.1 | 8.4 |
| $\text{HRA}_{r=32,\lambda=\infty}$ | 0.12% | 52.8 | 9.2 |
| $\text{HRA}_{r=32,\lambda=1e-1}$ | 0.12% | 53.6 | 8.3 |
| $\text{HRA}_{r=32,\lambda=1e-4}$ | 0.12% | 56.3 | 9.3 |
| $\text{HRA}_{r=32,\lambda=1e-8}$ | 0.12% | 53.6 | 8.6 |
| $\text{HRA}_{r=32,\lambda=0}$ | 0.12% | 55.8 | 9.0 |

We can find that a) the performance of HRA is relatively stable concerning the change of $\lambda$, b) in the wide range of $\lambda$, HRA is superior to the baselines, c) even if ignoring the regularizer $(\lambda=0$), our method still outperforms the baselines. These results demonstrate the effectiveness and robustness of implementing orthogonal adaptation based on Householder reflections. In the future, we will consider further analyzing the impacts of $\lambda$ in theory.

Regarding the societal impact of our work, we believe HRA can further simplify the adaptation of LLMs and promote more LLM-based downstream applications. Similar to LoRA and OFT, HRA may suffer from some potential issues like inappropriate (even illegal) abuse, amplifying the social prejudice intrinsically in LLM when the fine-tuning data are biased, and so on. **It should be noted that these potential issues are neither purely attributed to the technique itself nor specific to HRA—LoRA and OFT also suffer from them. Solving these issues depends on developing new techniques, social policies, and data quality improvement.** How to mitigate (even eliminate) these issues is left to our future work.

We will add the above content to the final version of our paper and the attached NeurIPS paper checklist.

**Q2: The comparisons for various methods on their training time and memory costs.**

**A2:** Following the suggestions of Reviewers 1vEG and mtnZ, we adapt LLaMA2-7B on the MetaMathQA dataset by HRA and other baselines and test their training time and GPU memory costs. For a fair comparison, we conducted all the experiments on 8 NVIDIA RTX A6000 GPUs, and all the methods applied the same batch size and almost the same number of trainable parameters.

| Method | #Param | Training time (hours) | Peak memory usage (GB) | GSM8K | MATH |
| :-----| :----: | :----: | :----: | :----: | :----: |
| LoRA | 0.12% | 45 | 279 | 47.3 | 6.6 |
| OFT | 0.13% | 53 | 282 | 50.1 | 8.4 |
| HRA | 0.12% | 30 | 287 | 56.3 | 9.3 |

We can find that HRA's peak memory usage is comparable to that of baselines, while its training time is less, and its accuracy in the downstream GSM8K and MATH tasks is better. These results demonstrate HRA's superiority in computational efficiency and adaptation performance.

We hope that the above response can resolve your concerns. Thanks again for your positive feedback.

---

### Decision · Program_Chairs · 2024-09-25

**Decision:**

Accept (spotlight)

**Comment:**

This paper presents an interesting idea based on Householder reflections to bridge the gap between two prevalent fine-tune methods, low-rank adaptation and orthogonal adaptation. After author-review discussions, the reviewers have reached consensus that the contribution of the paper is solid and significant, unanimously recommending accepting the paper. I would strongly recommend accepting the paper.